# The Endothelium as a Target for Anti-Atherogenic Therapy: A Focus on the Epigenetic Enzymes EZH2 and SIRT1

**DOI:** 10.3390/jpm11020103

**Published:** 2021-02-05

**Authors:** Jolien Fledderus, Byambasuren Vanchin, Marianne G. Rots, Guido Krenning

**Affiliations:** 1Medical Biology Section, Laboratory for Cardiovascular Regenerative Medicine, Department Pathology and Medical Biology, University Medical Center Groningen, University of Groningen, Hanzeplein 1 (EA11), 9713 GZ Groningen, The Netherlands; j.fledderus01@umcg.nl (J.F.); byambasuren.v@mnums.edu.mn (B.V.); 2Department Cardiology, School of Medicine, Mongolian National University of Medical Sciences, Jamyan St 3, Ulaanbaatar 14210, Mongolia; 3Epigenetic Editing, Medical Biology Section, Department Pathology and Medical Biology, University Medical Center Groningen, University of Groningen, Hanzeplein 1 (EA11), 9713 GZ Groningen, The Netherlands; m.g.rots@umcg.nl

**Keywords:** endothelial cell, endothelial dysfunction, atherosclerosis, arteriosclerosis, epigenetics, EZH2, SIRT1

## Abstract

Endothelial cell inflammatory activation and dysfunction are key events in the pathophysiology of atherosclerosis, and are associated with an elevated risk of cardiovascular events. Yet, therapies specifically targeting the endothelium and atherosclerosis are lacking. Here, we review how endothelial behaviour affects atherogenesis and pose that the endothelium may be an efficacious cellular target for antiatherogenic therapies. We discuss the contribution of endothelial inflammatory activation and dysfunction to atherogenesis and postulate that the dysregulation of specific epigenetic enzymes, EZH2 and SIRT1, aggravate endothelial dysfunction in a pleiotropic fashion. Moreover, we propose that commercially available drugs are available to clinically explore this postulation.

## 1. Introduction

Atherosclerosis is the leading cause of cardiovascular diseases and underlies pathologies such as heart attack, stroke, and peripheral vascular disease [1]. Atherosclerosis is a progressive inflammatory vascular disease, characterized by the thickening and hardening of the artery walls and the formation of atherosclerotic plaques, wherein lipids, extracellular matrix, immune cells, smooth muscle cells (SMCs), and (myo)fibroblasts accumulate [2]. The plaque progressively hardens and narrows the arteries, impairing blood flow. The rupture of unstable plaques may lead to thrombosis, one of the major clinical complications of atherosclerosis [3,4,5].

It is well established that endothelial cells (ECs) are majorly important in the maintenance of vascular homeostasis through the regulation of processes including vascular permeability, vascular tone, inflammation, vascular smooth muscle cell (VSMC) proliferation and thrombogenesis (reviewed in [6]). Endothelial dysfunction, the pathological state wherein a disbalance in these processes occurs, contributes to atherosclerosis development (reviewed in [7,8,9,10]). For example, oscillatory shear stress (OSS), as seen in aortic bifurcations, increases the susceptibility to atherosclerosis by the downregulation of ERK5, a kinase which evokes nitric oxide (NO) production and inhibits leukocyte–endothelial adhesion [11]. In mice lacking ERK5, atherosclerotic plaque formation is aggravated [12,13]. Corroboratively, alleviation of EC dysfunction reduces atherogenic plaque development [14,15], and may even evoke the regression of preestablished plaques [16].

These observations underscore the importance of endothelial dysfunction in the pathogenesis and progression of atherosclerosis and suggest that the endothelium may be an efficacious cellular target for antiatherogenic therapies. Remarkably, the clinically available therapies for atherosclerosis are symptomatic and focus on lowering circulating lipid levels, reducing the inflammatory processes, and the prevention of blood clotting [17]. Interestingly, no therapy is available that targets the dysfunctional endothelium. Here, we discuss the contribution of endothelial dysfunction to atherosclerosis development and progression, and postulate that the dysregulation of specific epigenetic enzymes aggravates endothelial dysfunction in a pleiotropic fashion. Moreover, we propose that commercially available drugs are available to clinically explore this postulation.

## 2. The Atherosclerotic Endothelium

The endothelium forms the innermost layer of all blood vessels. The healthy quiescent endothelium mediates vascular homeostasis by the inhibition of unwarranted inflammation, blood clotting, vasoconstriction, and the maintenance of the vascular barrier. Endothelial dysfunction manifests in lesion-prone areas of the vasculature and refers to a pro-inflammatory, prothrombotic and vasoconstrictive state of the endothelium wherein vascular permeability is often increased [7,8,9,10]. Endothelial dysfunction contributes to the development of atherosclerosis, which is exemplified by the clinical observation that atherosclerotic cardiovascular events occur at a higher frequency in patients with severe endothelial dysfunction than in patients with mild or moderated endothelial dysfunction [18]. Indeed, most atherosclerosis risk factors activate the endothelium and may induce endothelial dysfunction [10]. In this section, we discuss the mechanisms by which endothelial dysfunction contributes to the pathophysiology of atherosclerosis.

### 2.1. Endothelial Cell Activation, Dysfunction and the Formation of a Fatty Streak

Endothelial activation, triggered by atherosclerosis risk factors including oxidized lipids and inflammatory cytokines, is generally regarded as the first stage of atherogenesis (Figure 1). Endothelial cell activation results in the loss of intercellular junctions [19] and therefore affects the permeability of the vessel wall to infiltrating macromolecules such as low-density lipoprotein cholesterol (LDL-C). In the vessel wall, LDL-C undergoes oxidative modifications via its interaction with reactive oxygen species (ROS), and it is well recognized that this oxidized LDL-C (oxLDL) is pro-inflammatory and a major trigger of atherogenesis (reviewed in [20,21]).

Endothelial cells contribute to the increase in ROS in the vessel wall. In the healthy endothelium, the vasodilator nitric oxide (NO) is synthesized by the enzyme endothelial NO synthase (eNOS) [22,23] and endothelial cell-derived NO prevents the oxidation of LDL-C [7]. However, in activated endothelial cells, the production of NO is suppressed [24], resulting in increased vasoconstriction and an increase in the oxidation of LDL-C. The suppression of NO biosynthesis and LDL-C oxidation is perpetuated in the fatty streak, as oxLDL itself is a potent suppressor of NO synthesis [25]. The resulting “endothelial dysfunction” manifests as the earliest sign of atherogenesis and occurs even before structural changes in the vessel wall are apparent [7,26].

### 2.2. Inflammatory Cell Infiltration and Inflammation

Inflammation and inflammatory cell infiltration in the vessel wall are well-known contributors to atherosclerosis development, progression and regulators of plaque stability (reviewed [27,28]) (Figure 1). The accumulation of oxLDL in the fatty streak (discussed above), induces the production of a variety of inflammatory molecules by the endothelium, including adhesion molecules (e.g., E-Selectin, VCAM1 and ICAM1), chemotactic proteins (e.g., monocyte chemoattractant protein 1 (MCP-1)) and pro-inflammatory cytokines (e.g., IL-1β, TNFα, macrophage colony-stimulating factor (M-CSF)) [29,30].

Interestingly, atherosclerosis susceptibility in mice critically depends on this pro-inflammatory response by the endothelium [31].

The induction of endothelial adhesion molecules expression facilitates the invasion of inflammatory cells into the forming atherosclerotic plaque [32]. For example, E-selectin mediates the rolling of leukocytes along the endothelial surface, whereas VCAM-1 and ICAM-1 mediate firm adhesion of leukocytes to the endothelium which is required for extravasation. Indeed, in atherosclerosis prone, VCAM1-deficient mice (ApoE^−/−^;VCAM1^D4D/D4D^), fatty streak formation and atherogenesis is blunted when compared to atherosclerosis-prone mice that do express VCAM1 (ApoE^−/−^VCAM1^+/+^ mice) [33].

In addition, endothelial cell activation results in the expression of inflammatory cell chemoattractants and mitogens, including MCP-1 and M-CSF. MCP-1 recruits monocytes into the vessel wall, whereas M-CSF induces monocyte proliferation and differentiation into macrophages [34]. In the vessel wall, macrophages phagocytize oxLDL and form foam cells, which aggregate to form an atheromatous core [35], and perpetuates atherogenesis by the amplification of ROS and inflammatory cytokine production [9,36].

Besides the direct pro-inflammatory effects of oxLDL on the endothelium, persistent inflammatory signalling in the atherosclerotic milieu can induce cellular senescence. Cellular senescence is the phenomenon where cells cease to divide in response to telomere shortening or biochemical stress (e.g., ROS accumulation and DNA damage) [37]. Senescent endothelial cells adopt a pro-inflammatory and pro-thrombotic phenotype, also known as the senescence-associated secretory phenotype (SASP). Multiple molecules, including growth factors, cytokines and metalloproteinases, constitute the SASP, which can all promote atherogenesis [38]. Senescent ECs are found at sites of atherogenesis and contribute to the increased vascular permeability, persistent inflammation and ongoing vascular remodelling [39,40].

### 2.3. Neointima Formation

Intima hyperplasia and neointima formation are the third stage of atherogenesis. This stage is characterized by the formation of a fibrous plaque consisting of accumulating VSMC and inflammatory cells, and extracellular matrix (ECM) in response to the inflammatory milieu (Figure 1). VSMC migrate from the vascular media towards the neointima on a gradient of pro-inflammatory cytokines secreted by T cells, macrophages, and foam cells. Recruited VSMC in turn secrete fibrous ECM proteins that contribute to the increase and stiffening of the atherosclerotic plaque [7].

ECs also regulate the proliferation and migration of VSMC in the atherosclerotic plaque. EC-derived NO inhibits the proliferation of VSMC, whereas EC-derived factors such as Endothelin-1 (ET-1) and Angiotensin II (AT2) promote the proliferation of VSMC [41]. As mentioned above, activated ECs at the site of atherogenesis produce less NO, causing a relative shift in the availability of VSMC mitotic factors.

ECs may also directly contribute to neointima formation via the process of mndothelial–mesenchymal transition (EndMT), in which ECs transform into apoptosis-resistant, fibroproliferative mesenchymal-like cells that accumulate in the neointima and the fibrous cap of the atherosclerotic lesion [10,42,43,44]. EndMT may contribute to increased leukocyte extravasation, intimal lipid accumulation, ECM accumulation and oxidative stress in the atherosclerotic plaque [13,16,45,46]. It is increasingly recognized in the field that metastable or partial EndMT occurs in atherosclerosis, in which ECs maintain some of their original endothelial functions, yet express inflammatory cytokines and an abundance of ECM molecules [47].

### 2.4. Thrombogenesis

The final stage and major complication of atherosclerosis is thrombosis (Figure 1). Thrombosis occurs as a consequence of atherosclerotic plaque rupture and its occurrence critically depends on the vulnerability of a plaque which is determined by the thickness of the fibrous cap [9,36]. More precisely, vulnerable plaques have thin fibrous caps, which are low in VSMC numbers and have a high load of inflammatory cells. The thinning of the cap is promoted by VSMC senescence or death and degradation of the cap’s ECM proteins [48,49,50,51]. Indeed, the proliferation rate of VSMC is higher in early and stable lesions when compared to the proliferation rate of advanced and vulnerable atherosclerotic plaques [52,53,54], suggesting that VSMC senescence may contribute to plaque rupture.

Endothelial cells may also contribute to plaque vulnerability and the concurrent thrombosis via EndMT [46]. Although EndMT increases the number of mesenchymal cells in the fibrous cap, plaque vulnerability counterintuitively associates with increased EndMT, suggesting that the secretome of cells undergoing EndMT may promote ECM destabilization. Indeed, EndMT-derived mesenchymal cells in the fibrous cap secrete various ECM-degrading proteins in high amounts [46] resulting in plaque destabilization.

From the above, it becomes apparent that endothelial damage (i.e., the sum of endothelial activation, endothelial dysfunction, senescence and (partial) EndMT) pivotally contributes to atherogenesis at every stage of pathogenesis (Figure 1). We therefore postulate that the endothelium may serve as an efficacious therapeutic target cell for anti-atherogenic therapies. In the next sections, we review the current available anti-atherogenic therapies and discuss the potential to restore endothelial homeostasis using epigenetic drugs that may alleviate endothelial damage at multiple levels.

## 3. Current and Experimental Atherosclerosis Therapies

Current medical treatments to prevent atherosclerosis development, progression or plaque rupture include anti-hypertensive and lipid-lowering drugs to reduce atherosclerotic risk burden and anti-thrombogenic therapies to limit the complications of atherosclerosis, whereas experimental therapies primarily focus on anti-inflammatory agents to reduce atherosclerosis progression (Figure 2). In this section, we discuss the currently available clinical and experimental medicaments and their rationale as an anti-atherogenic agent (Table 1).

### 3.1. Antihypertensive and Lipid-Lowering Drugs

Hypertension is one of the primary risk factors for atherosclerosis. Hypertension accelerates the atherogenesis and a persisting high blood pressure increases plaque vulnerability [55] and the associated risk of major cardiovascular events. Thus, anti-hypertensive medicaments are prescribed to patients unable to lower their blood pressure by dietary- or lifestyle adaptations [56]. Competitive antagonists of the adrenergic beta receptors (commonly known as β-blockers), are inhibitors of the angiotensin-converting enzyme (ACEi), calcium channel blockers and reduce blood pressure and slow the atherogenic process [57,58,59].

Dyslipidaemia is another primary risk factor for atherosclerosis and lipid-lowering interventions are the principle pharmaceutical intervention for the treatment of atherosclerosis at present [63,74]. Lipid-lowering interventions aim to reduce the residual lipid risk, as epidemiological cohort studies and clinical trials imply that elevated levels of LDL-C and oxLDL strongly associate with atherogenesis and cardiovascular events (reviewed in [75]). β-hydroxy-β-methylglutaryl-CoA (HMG-CoA) reductase inhibitors (commonly known as statins) reduce the production of LDL-C and facilitate LDL-C clearance by increasing LDL receptor expression in the liver [76]. Besides these primary effects, statins are anti-inflammatory [77,78] and increase endothelial NO production through the activation of eNOS [77,79], which may alleviate endothelial dysfunction during atherosclerosis. This pleiotropy of statins may explain why statins outperform other lipid-lowering drugs and are the first-choice medicament for secondary prevention of atherosclerosis. Other commonly prescribed lipid-lowering drugs include the proprotein convertase subtilisin/kexin type 9 (PCSK9) inhibitors and cholesterol absorption inhibitors, which both increase LDL-C clearance from the blood [80,81]. Interestingly, and given that lipid-lowering drugs reduce the incidence of severe cardiovascular events [62,82], the sole treatment of dyslipidaemia does not prevent atherogenesis [83,84], underscoring the need for additional therapies.

### 3.2. Anti-Inflammatory Agents

Chronic low-grade systemic inflammation aggravates atherogenesis [27,28], suggesting that anti-inflammatory agents may be beneficial in atherosclerosis treatment. Inflammatory signalling is involved in atherogenic processes (e.g., EC activation [85,86], inflammatory cell invasion [86,87], foam cell formation [88,89], senescence [90] and VSMC migration and proliferation [91,92]), and the identification of druggable anti-inflammatory targets is the subject of intense research in cardiovascular medicine. Monoclonal antibodies raised against inflammatory cytokines (e.g., canakinumab that targets IL-1β) reduce inflammatory signalling in multiple cell types and may therefore have a broad anti-atherogenic effect [64]. Indeed, atherosclerosis patients experimentally treated with canakinumab have a lowered risk for future severe cardiovascular outcomes [65]. Agents impacting alternative inflammatory pathways include the targeting of leukotrienes (e.g., atreleuton that targets 5-lipoxygenase [67]), chemokines (e.g., C-C motif chemokine ligand 2 (CCL2)-C-C chemokine receptor type 2 (CCR2) inhibitors [71]), and adhesion molecules (e.g., inclacumab that targets P-selectin [66]), which all aim to mitigate inflammatory cell recruitment into the atherosclerotic plaque [93,94,95]. Finally, phospholipase A inhibitors (e.g., darapladib that targets lipoprotein-associated phospholipase A2 [69]) mitigate the formation of pro-atherogenic lipid moieties that attract inflammatory cells and cause foam cell differentiation.

### 3.3. Anti-Thrombotic Agents

Atherothrombosis occurs when an atherosclerotic plaque ruptures, and is the main cause of atherosclerosis-associated mortality [96]. The healthy quiescent endothelium produces a variety of anti-thrombogenic actors, including prostacyclin [97], anti-thrombin [98], thrombomodulin [99], and NO [100]. Yet, following plaque rupture, the damaged endothelium cannot maintain haemostasis and thrombogenesis initiates. Hence, anti-aggregants (i.e., agents that mitigate platelet aggregation) and anticoagulants (i.e., agents that inhibit blood clotting) are commonly prescribed as prophylaxis for cardiovascular events at late stages of atherosclerosis [101].

To summarize, current clinical and experimental antiatherogenic therapies are symptomatic and target hypertension, dyslipidaemia, and inflammation. Even though the endothelium may represent an efficacious cellular target for antiatherogenic therapies and endothelial damage is present throughout atherogenesis, remarkably, no therapies are in development that focus specifically on resolving endothelial damage to stop or even revert atherosclerosis progression. In the following section, we discuss the benefits of endothelial-specific antiatherogenic therapies, with a focus on epigenetic interventions, as these may broadly affect endothelial behaviour and may reverse dysfunction originating from multiple mechanisms.

## 4. The Endothelial Transcriptome as a Target for Anti-Atherogenic Therapy

In atherogenesis, endothelial damage occurs as a consequence of transcriptional changes in response to atherogenic stimuli [102,103], and the responsiveness of the endothelium to these stimuli determines the genetic atherosclerosis susceptibility in mice [31,104]. Moreover, the endothelial transcriptome differs between atheroprotective and atheroprone areas of the same vessel [105,106].

Determinants of the endothelial transcriptome are multiple, including the presence and activity of transcription factors (TFs), epigenetic regulation of the accessibility of the DNA to TFs, ribosomal activity, and post-transcriptional regulation via non-coding RNAs. The adaptation of a proatherogenic transcriptome by ECs may thus be the consequence of any of these factors alone or in combination, and implies that their therapeutic targeting may provide therapeutic benefit. As a single epigenetic enzyme may affect the endothelial transcriptome more broadly than any other singular factor, in the following section, we discuss the potential of small molecule agonists and antagonists of such enzymes to “normalize” the endothelial transcriptome to mitigate atherogenesis.

### 4.1. Epigenetic Regulation of the Endothelial Pro-Atherogenic Phenotype

Epigenetics refers to heritable phenotypical changes that do not involve changes in the genome, and include DNA and histone modifications [107]. Core histone proteins contain a globular domain and an amino terminal tail which can be modified by—amongst others—acetylation, methylation, phosphorylation, and sumoylation [108], that modify gene expression by modifying chromatin structure and regulating the accessibility of the chromatin to TFs.

Research into epigenetic changes during atherogenesis is in its infancy, yet pioneering studies are now being reported. In early atherogenesis, genome-wide acetylation of lysine (K) 9 from histone (H) 3 and H3K27 is increased, whereas methylation (Me) of H3K4 and H3K9 is observed during late atherogenesis [109]. Interestingly, H3K27Me3 is decreased in VSMCs [109] and increased in ECs during atherogenesis [110], suggesting cell type-specific epigenetic regulation within a similar extracellular milieu. Nonetheless, increased expression of enhancer of zeste homologue 2 (EZH2)—the epigenetic enzyme that places the trimethylation mark on H3K27—promotes atherogenesis in mice [111].

Endothelial cell damage and dysfunction is similarly affected by epigenetic modifications; the histone deacetylase sirtuin 1 (SIRT1) mitigates hyperglycaemia-induced EC dysfunction [112], the methyltransferase SUV39H1 mitigates endothelial oxidative stress [113], and endothelial inflammatory activation can be abrogated by blunting the expression of the methyltransferase Set7 [114]. So far, two epigenetic enzymes, namely the histone methyltransferase EZH2 and the histone deacetylase SIRT1, have been convincingly described to affect endothelial cell function at multiple levels (Table 2), and it is tempting to speculate that the therapeutic targeting of these epigenetic enzymes may limit the development or progression of atherogenesis. In the following section, we discuss how these two epigenetic enzymes affect endothelial cell functions and how their therapeutic targeting changes the outcome of atherogenesis.

### 4.2. Endothelial Enhancer of Zeste Homologue 2 (EZH2)

Enhancer of zeste homologue 2 (EZH2) is a methyltransferase that trimethylates lysine 27 of histone 3 (H3K27Me3) and affects many endothelial cell functions [115,116]. As methyltransferase, EZH2 is the catalytic subunit of the polycomb repressive complex 2 (PRC2) that silences gene expression through chromatin compaction [117] thereby making the DNA inaccessible for the transcription machinery [118].

In the context of atherosclerosis, hyperhomocysteinemia—a well-recognized risk factor for atherogenesis [143]—associates with elevated H3K27Me3 abundance, decreased NO production, EC apoptosis and an increase in lesion fat accumulation and lesion size [144,145,146,147]. Likewise, LDL-C induces endothelial EZH2 expression and mitigates KLF2-dependent NO production, resulting in vasoconstriction and the decreased expression of anti-atherogenic factors thrombomodulin and plasminogen activator inhibitor-1 (PAI-1) [119]. Moreover, in areas with OSS, ECs have increased H3K27Me3 abundance [110,116] and via the downregulation of ERK5 and consequently KLF2 and KLF4, two transcription factors that drive the expression of atheroprotective genes, this associates with a highly pro-inflammatory, pro-apoptotic and procoagulant endothelial [115,148,149,150,151]. Expectedly, RNAi-mediated silencing of EZH2 in the endothelium decreased pro-inflammatory gene expression, and thus reduces inflammatory cell recruitment, foam cell formation and atherogenesis [110]. An example of the beneficial effects of EZH2 silencing comes from the field of oncology. Epithelial–mesenchymal transition, a process similar to EndMT, is involved in several types of cancer and is promoted by EZH2 via the upregulation of the transcription factor Snail [152,153,154]. Therefore, pharmacological targeting of EZH2 inhibits epithelial–mesenchymal transition [155]. In the context of atherosclerosis where EndMT is also associated with an upregulation of Snail, this indicates that the repression of EZH2 could reduce EndMT and therefore, contributes to the healthy endothelium.

The molecular mechanisms by which EZH2 may affect endothelial-dependent atherogenesis are gaining attention. The independent atherosclerosis risk factors dyslipidaemia [130], hyperhomocysteinemia, [144] hyperglycaemia [156] and OSS [110] all block the expression of endothelial microRNAs that repress EZH2 translation under physiological conditions. For example, in human umbilical vein endothelial cells (HUVECs) treated with oxLDL, miR-200a expression is decreased, causing an upregulation of EZH2 [130]. During hyperhomocysteinemia, levels of EZH2 are increased as a result of reduced levels of miR-92a [144] and upon OSS, miR-101 expression is downregulated [110,157] also leading to an increased expression of endothelial EZH2 and the concurrent H3K27Me3 [130,144,156]. Interestingly, in the field of oncology, multiple other microRNAs such as miR-26a, miR-124 and miR-214 have been described to repress EZH2 expression [158,159]. This could imply a role for these microRNAs in the field of atherogenesis.

Next, long non-coding RNAs (lncRNAs) guide EZH2 towards target genes [120], where critical endothelial genes are silenced. This orchestrated mechanism ultimately culminates in endothelial inflammatory activation, dysfunction and cell death [120,130,144,156].

### 4.3. Endothelial NAD^+^-Dependent Deacetylase Sirtuin 1 (SIRT1)

Sirtuin 1 (SIRT1) is a NAD^+^-dependent deacetylase that removes acetyl groups from histone tails with limited selectivity. SIRT1 facilitates gene silencing by the removal of acetyl groups from activating histone marks, including H1K26Ac, H3K9Ac, and H4K16Ac [160]. Interestingly, the loss of SIRT1 activity aggravates atherogenesis in ApoE^−/−^ mice and the pharmacological activation of SIRT1 decreases atherosclerotic plaque size [161].

In the context of atherogenesis, the endothelial cell-specific overexpression of SIRT1 blunts atherogenesis in hypercholesteraemic ApoE^−/−^ mice [128]. SIRT1 decreases the endothelial reactivity to oxLDL [122,124,125], potentially by limiting its cellular uptake [121]. Furthermore, SIRT1 increases the endothelial production of NO by increasing the expression and the activity of the NO producing enzyme eNOS [122,127,162]. As discussed above, the bioavailability of NO may influence atherogenesis at multiple levels [163].

SIRT1 also inhibits the inflammatory activation of endothelial cells via several distinct mechanisms. First, SIRT1 can deacetylate the p65 subunit of the inflammatory transcription factor Nuclear Factor Kappa B (NFκB), resulting in its deactivation [123,164], and thus blunting the endothelial inflammatory response [131]. Second, SIRT1 decreases inflammatory activation by inhibiting the p66shc–ROS–NFκB axis. P66shc is an oxLDL-sensitive gene that activates membrane bound NADPH oxidases (Nox) in ECs and deactivates ROS detoxifying enzymes, culminating in superoxide stress [165]. SIRT1 reduces p66shc expression by the deacetylation of histone 3 in its promoter region [112], resulting in the suppression of Nox-dependent [122,125] and mitochondrial [124] superoxide production. Third, SIRT1 reduces the formation of NLRP3 inflammasomes and the concurrent secretion of pro-inflammatory cytokines through a non-elucidated mechanism [121]. Fourth, SIRT1 precludes oxidative DNA damage and the concurrent induction of endothelial cell senescence [133], through a mechanism that involves the deacetylation of the p53 transcription factor [134]. Thus, these data collectively attribute that SIRT1 averts endothelial inflammatory activation and senescence and may thereby attenuate atherogenesis.

Next to its role in maintaining endothelial function and precluding endothelial inflammatory activation, SIRT1 may also attenuate the endothelial contribution to neointima formation by regulating EndMT. Albeit the mechanism by which SIRT1 blocks EndMT is incompletely understood: SIRT1 blunts TGFβ signalling and EndMT through the inhibition of SMAD2/3 nuclear translocation and the deacetylation of SMAD4 [137,138]. Besides limiting the EndMT-derived neointimal cell number, SIRT1 also inhibits the production of an endothelial fibrogenic secretome, which may attenuate VSMC proliferation in the neointima [139].

Last, SIRT1 modulates endothelial anti-thrombogenicity [141]. SIRT1 deacetylates H4K16 in the PAI-1 promoter [141], which precludes its expression under atherogenic stress [132,136,140]. Furthermore, SIRT1 activation maintains thrombomodulin activity [142], suggesting SIRT1 evokes thrombolysis in ECs and may prevent clotting during atherogenesis.

### 4.4. EZH2 and SIRT1: the Yin and Yang of Early Atherogenesis

As discussed above, both EZH2 antagonism as well as SIRT1 agonism, have promising endothelial cell-mediated antiatherogenic effects (Figure 3). However, it should be noted here, that EZH2 inhibition and SIRT1 activation also have anti-atherogenic effects that are endothelial cell independent. For instance, inhibition of EZH2 as a result of GAS5 knockdown, a lncRNA that directly interacts with EZH2, promotes ABCA1-dependent hepatic LDL-C clearance [166]. Also, EZH2 inhibition mitigates foam cell oxLDL uptake in ApoE^−/−^ mice [121]. Similarly, SIRT1 activation limits hepatic LDL-C production [161] and reduces foam cell formation [167] in atherosclerotic mice. Nonetheless, the endothelial-derived beneficial effects evoked by these interventions may greatly attenuate the atherogenic process.

Interestingly, and notwithstanding the individual atheroprotective effects evoked by EZH2 inhibition or SIRT1 activation alone (Table 2), target genes for EZH2 and SIRT1 overlap [168], suggesting an interconnection between the effects these epigenetic enzymes evoke during atherogenesis, which might suggest therapeutic synergy when interventions at the level of EZH2 and SIRT1 are combined. By targeting an epigenetic effector to a given gene, using epigenetic editing may be uniquely suited to unravel such seemingly interconnected mechanisms. Interestingly, as gene expression changes can be epigenetically induced in a sustained manner [169,170], this approach may also bear therapeutic opportunities.

## 5. Future Clinical Perspective

From the previous, it becomes visible that the pharmacological inhibition of EZH2 and activation of SIRT1 may be promising anti-atherogenic targets by mitigating endothelial dysfunction. The discovery of “epigenetic drugs” (i.e., small molecules that target epigenetic enzymes) is a field of active research, and new therapeutics are entering the clinic [171]. These currently available “epigenetic drugs” affect the epigenome genome-wide and may not show full selectivity, which may evoke unwanted side-effects. Yet, if a gene-specific causal epigenetic modification in known, epigenetic editing, i.e., the targeted modification of a specific epigenetic mark, may offer new solutions [172]. Exciting developments in this field show that gene expression can be specifically and sustainably induced through the epigenetic editing of DNA methylation [170] and histone modifications [169].

Epigenetic drugs that antagonize EZH2 and agonize SIRT1 activity are already clinically available or in development [171], albeit for indications other than atherosclerosis (Table 3). The efficacy of the EZH2 antagonist tazemetostat is currently being assessed in oncology, implying that this antagonist is safe for human use. The safety profile of EZH2 antagonists GSK2816126 and SHR2554 is currently assessed in phase I trials. In the experimental setting, tazemetostat decreases endothelial inflammatory signalling [129], but cardiovascular data on GSK2816126 and SHR2554 are lacking. As cancer development and atherogenesis seem to share an overlapping epidemiology and have a high co-occurrence [173], the inclusion of cardiovascular secondary endpoints in these clinical studies would facilitate a fast assessment of the antiatherogenic potential of EZH2 antagonists.

SIRT1 agonists have received more clinical attention in the cardiovascular field, and several SIRT1 agonists—including resveratrol, quercetin, and SRT2104—are currently being examined in clinical studies (Table 3). In the experimental setting, resveratrol precludes vascular inflammation and attenuates atherogenesis [162,174,175]. Quercetin mitigates endothelial cell activation, inflammation, VSMC proliferation, and lowers LDL-C uptake [176,177,178,179]. SRT2104 (also known as SRT3025) reduces endothelial oxidative stress and senescence, and mitigates atherogenesis via increased LDL-C excretion [135,161,180]. Clinically, resveratrol, quercetin and SRT2104 all improve the serum lipid profile [181,182,183], which supports further development as an anti-atherogenic drug.

Thus, compounds directed at normalizing EZH2 and SIRT1 activity can promote endothelial homeostasis, and thereby limit atherogenesis. Compounds under development for alternative indications can readily serve to explore the clinical potential for vascular health. To test the effect on atherogenesis, non-invasive cardiac endpoints, such as carotid Doppler ultrasound, might be employed in ongoing trials, allowing the early assessment of therapeutic potential in the cardiovascular field. If promising, add-on studies can be performed in patients suffering from atherosclerosis. Interestingly SIRT1 activity in circulating mononuclear cells is associated with atherosclerosis risk [184] and may serve as an accessible biomarker for patient inclusion in such studies.

## 6. Conclusions

Cardiovascular diseases remain the number one cause of death worldwide, and atherosclerosis is the leading pathology. Endothelial inflammatory activation and dysfunction are critical components in the development and progression of atherosclerosis by contributing to fatty streak formation, vascular inflammation, and neointima formation. Remarkably, current anti-atherogenic therapies do not include interventions at the endothelial level. We argue that endothelial cell EZH2 and SIRT-1 are involved in the regulation of endothelial homeostasis and their dysregulation contributes majorly to atherogenesis. Validation of driver genes and the clinical development of the epigenetic editing platforms would open future innovative avenues to prevent or treat atherosclerosis. Currently, compounds that antagonize EZH2 or agonize SIRT1 are clinically available or in development and warrant the inclusion of cardiovascular endpoints in the respective ongoing clinical studies. Insights into the effects of epigenetic drugs will indicate the role of epigenetics and facilitate the establishment of epigenetic therapies as anti-atherogenic medicaments.

## Figures and Tables

**Figure 1 jpm-11-00103-f001:**
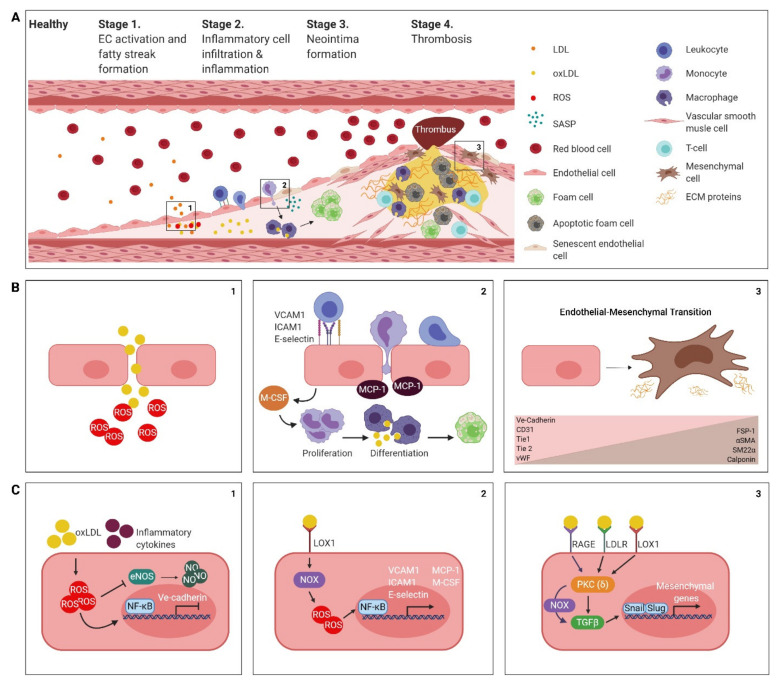
Endothelial cell behaviour during atherosclerotic plaque formation. (**A**) Atherosclerosis risk factors evoke endothelial cell activation, oxidative stress, and dysfunction in endothelial cells, resulting in increased vascular permeability to lipids (1), and the infiltration of inflammatory cells into the vessel wall (2). Endothelial cells (ECs) may also contribute to neointima formation by endothelial–mesenchymal transition (3). (**B**) Detailed view of intercellular interactions during (1) increased vessel permeability, (2) vessel wall inflammation and foam cell differentiation and (3) endothelial–mesenchymal transition. Loss of EC markers are represented in pink and gain of mesenchymal markers are represented in brown. (**C**) Overview of intracellular signalling in endothelial cells that links dyslipidaemia to (1) endothelial cell oxidative stress and dysfunction, (2) inflammatory activation, and (3) endothelial-mesenchymal transition. (ox)LDL: (oxidized) low density lipoprotein, ROS: reactive oxygen species, SASP: senescence-associated secretory phenotype, ECM proteins: extracellular matrix proteins, MCP-1: monocyte chemoattractant protein 1, M-CSF: macrophage colony-stimulating factor, eNOS: endothelial nitric oxide synthase, NO: nitric oxide.

**Figure 2 jpm-11-00103-f002:**
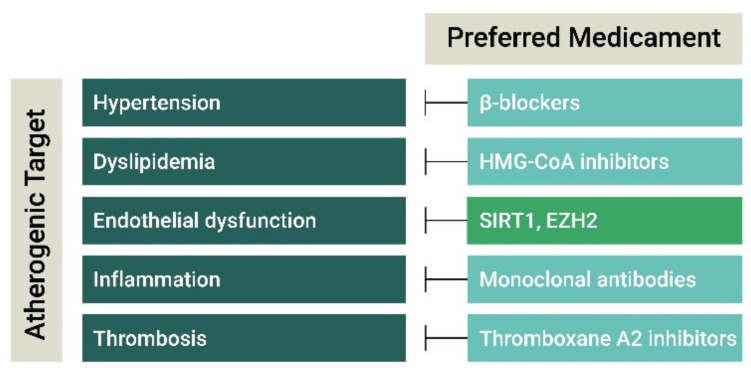
Current and future anti-atherogenic therapies. Established atherogenic risk factors and processes are depicted in green bars and the current available and experimental therapies to counteract these atherogenic processes are depicted in the neighbouring bars. Current anti-atherogenic therapies successfully preclude hypertension, dyslipidaemia, inflammation, and thrombosis, however, endothelial dysfunction is not specifically targeted therapeutically. In this review we explain that the therapeutic targeting of the epigenetic enzymes EZH2 and SIRT1 mitigates endothelial cell activation and dysfunction and may block atherogenesis.

**Figure 3 jpm-11-00103-f003:**
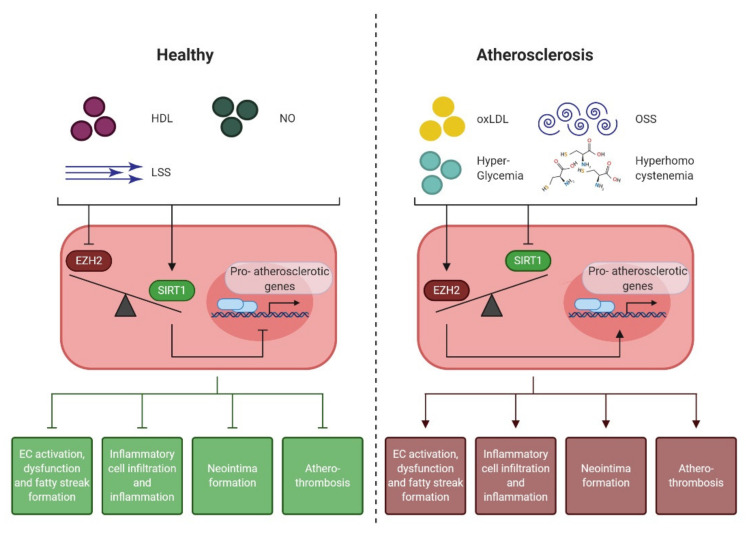
Endothelial cell-derived atheroprotective effects mediated by EZH2 and SIRT1. During homeostasis, high SIRT1, and low EZH2 activity ensure endothelial homeostasis, characterized by NO production, anti-inflammatory signalling, and the inhibition of the expression of atheroprone genes. When the balance between EZH2 and SIRT1 signalling is disturbed by atheroprone risk factors (e.g., oxLDL, OSS, glycemia or hyperhomocysteinemia), endothelial dysfunction is initiated, which accumulated to lipid streak formation, inflammatory cell infiltration and neointima formation. Therapeutic restoration of the balance between EZH2 and SIRT1 activity may provide a novel treatment to preclude atherogenesis. HDL: high density lipoprotein, NO: nitric oxide, LSS: laminar shear stress; (ox)LDL: (oxidized) low density lipoprotein, OSS: oscillatory shear stress.

**Table 1 jpm-11-00103-t001:** Current and experimental anti-atherogenic therapies.

Atherogenic Target	Class	Compound	Molecular Target	Stage of Development	Refs
Hypertension	β-blockers	Metoprolol, Carvedilol, Bisoprolol	adrenergic β-receptors	Marketed	[56,57,60,61]
	ACE inhibitors	Captopril, Benazepril, Perindopril, Ramipril	Angiotensin-converting enzyme		
	Ca^2+^-channel blockers	Amlodipine, Nifedipine	Voltage-dependent L^−^, N^−^, and T-type Ca^2+^ channels		
	Diuretics	Thiazide	Solute carrier family 12 members		
	Angiotensin-Receptor blockers	Losartan, Valsartan	Angiotensin receptor	Marketed	[60,61,62,63]
Dyslipidemia	HGM-CoA inhibitors	Statins	HMG-CoA	Marketed	[17,62]
	PCSK9 inhibitors	Evolucumab, Alirocumab	PCSK9		
	Cholesterol absorption inhibitors	Ezetimibe	NPC1L1, SOAT1		
Inflammation	Antibodies	Canakinumab; Adalimumab, Infliximab, Inclacumab	Cytokines (IL1β, TNFα), adhesion molecules (P-selectin)	I-III	[64,65,66]
	Lipoxygenase inhibitors	Atreleutron, Veliflapon	5-LO, FLAP	II	[67,68]
	Phospholipase inhibitors	Darapladib, Varespladib	Lp-PLA2, sPLA2	III	[69,70]
	CCL2-CCR2 inhibitors		CCR2	I	[71]
Thrombosis	Thromboxane A2 inhibitors	Aspirin	Cyclooxygenases	Marketed	[72,73]
	P2Y_12_ inhibitors	Clopidegril, Ticagrelor, Prasugrel, Cangrelor	P2Y purinoceptor 12		
	GPIIb/IIIa inhibitors	Tirofiban, Eptifibatide, Abciximb	platelet glycoprotein (GP) IIb/IIIa receptor		
	PAR-1 inhibitors	Vorapaxar	Proteinase-activated receptor 1		

5-LO = 5-lipoxygenase; CCL2 = C-C motif chemokine ligand 2 (monocyte chemoattractant protein (MCP)-1); CCR2 = C-C chemokine receptor type 2; FLAP = 5-LO activating protein; HMG-CoA = β-hydroxy-β-methylglutaryl-CoA; Lp-PLA2 = lipoprotein-associated phospholipase A2; NPC1L1 = Niemann–Pick C1-like protein 1; PSCK9 = proprotein convertase subtilisin/kexin type 9; SOAT1 = sterol O-acyltransferase 1; sPLA2 = secretory phospholipase A2.

**Table 2 jpm-11-00103-t002:** Endothelial cell-mediated antiatherogenic effects induced by EZH2 antagonism or SIRT1 agonism.

Atherogenic Phase	Endothelial Cell-Derived Atheroprotective Effects	
	EZH2 Antagonism	SIRT1 Agonism
Endothelial cell activation, dysfunction, and fatty streak formation	· EZH2 antagonism increases NO production [119] · EZH2 antagonism decreases ECs apoptosis [120]	· SIRT1 agonism decreases EC oxLDL uptake [121] · SIRT1 agonism decreases endothelial oxidative stress [121,122,123,124,125] · SIRT1 agonism increases NO production [126,127,128]
Inflammatory cell infiltration and inflammation	· EZH2 antagonism decreases EC inflammatory activation via increased ERK5 [110,115,116,129,130]	· SIRT1 agonism decreases EC inflammatory activation [112,131] · SIRT1 agonism decreases EC senescence [132,133,134,135,136]
Neointima formation	· EZH2 antagonism attenuates TGFβ-induced EndMT [116]	· SIRT1 agonism blocks TGFβ-induced EndMT [137,138] · SIRT1 agonism blocks endothelial secretion of fibrogenic factors [139]
Atherothrombosis		· SIRT1 agonism increases EC anti-thrombogenicity [140,141,142]

**Table 3 jpm-11-00103-t003:** Industry-sponsored active and completed clinical studies investigating EZH2 antagonism or SIRT1 agonism.

Mechanism	Drug	Field of Use	Clinical Phase of Development	Number of Active Studies	Developer	Clinical Trial Identifier (s)
EZH2 antagonist	CPI-0209	Oncology	I–II	1	Constellation Pharmaceuticals	NCT04104776
	CPI-1205	Oncology	I–II	3	Constellation Pharmaceuticals	NCT02395601, NCT03525795, NCT03480646
	GSK2816126	Oncology	I	1	GlaxoSmithKline	NCT02082977
	HH2853	Oncology	I	1	Haihe Pharmaceutical	NCT04390737
	PF-06821497	Oncology	I	1	Pfizer	NCT03460977
	SHR2554	Oncology	I–II	5	Jiangsu HengRui Medicine	NCT04577885, NCT04627129, NCT04335266, NCT03741712, NCT04407741
	MAK683		I	1	Novartis	NCT02900651
	Tazemetostat	Oncology	I–III	15	Epizyme	NCT03009344, NCT03010982, NCT03028103, NCT02220842, NCT02875548, NCT03456726, NCT01897571, NCT02889523, NCT03155620, NCT04204941, NCT04224493, NCT02860286, NCT04179864, NCT02601950, NCT03854474
SIRT1 agonist	Quercetin	Cardiovascular	II–III	3	Quercegen Pharmaceuticals, Boehringer Ingelheim	NCT03943459, NCT02195232, NCT02191280
		Orthopaedics	III	1	Nestlé	NCT00330096
		Pulmonology	I–II	3	Quercegen Pharmaceuticals, AB Science	NCT03989271, NCT01708278, NCT04622865
	Resveratrol	Cardiovascular	I–III	6	Atrium Innovations, DSM Nutritional Products, Gateway Health Alliance, KGK Science, Fluxome Sciences	NCT01964846, NCT01364961, NCT01564381, NCT02415114, NCT01914081, NCT01668836
		Dermatology	I	1	TCI Co	NCT04456829
		Metabolic	II–III	9	DSM Nutritional Products, Vedic Lifesciences	NCT01038089, NCT02216552, NCT02129595, NCT02565979, NCT01635114, NCT00823381, NCT00998504, NCT02834078, NCT02219906
		Neurology	I–II	9	Bial-Portela (BIA 6-512), Evolva (Veri-te)	NCT03095092, NCT03093389, NCT03095105, NCT03091543, NCT03094156, NCT03097211, NCT04314739, NCT03448094, NCT02621554
		Nephrology	III	1		NCT02433925
		Oncology	I		GlaxoSmithKline	NCT00920803
		Pulmonology	I–III	1	DSM Nutritional Products (Resvida)	NCT02245932, NCT02245962, NCT04166396
	SRT2379	Metabolic	I	2	GlaxoSmithKline	NCT01262911, NCT01416376
		Nephrology	I	1	GlaxoSmithKline	NCT01018628
	SRT2104	Metabolic	I–II	7	GlaxoSmithKline	NCT00938275, NCT00933530, NCT00933062, NCT00937872, NCT01031108, NCT00937326, NCT01018017
		Nephrology	I	1	GlaxoSmithKline	NCT01014117
		Pulmonology	I	1	GlaxoSmithKline	NCT00920660

Source: ClinicalTrials.gov.

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
