# Peer review of "The Endothelium as a Target for Anti-Atherogenic Therapy: A Focus on the Epigenetic Enzymes EZH2 and SIRT1"

_jpm, 2021, doi:10.3390/jpm11020103_

Round 1
Reviewer 1 Report
Manuscript written by Jolien Fledderus e.al., talks about the role of two epigenetic enzymes; EZH2 and SIRT1. The article is written nicely. However, authors have given less attention to EZH2 and SIRT1. Half of the article is talking about basic of atherosclerosis in detail. Only sections; 4 talks about these two epigenetic enzymes. My major comments on the article are;
The section 1 and 2 can be reduced since general process of atherosclerosis is already known and well-defined.
In section 4, role of SIRT1 is well established in cardiovascular complications. Break this section into two epigenetic enzymes as sub-sections. They cannot be discussed together. Discuss more about EZH2 since its mostly targeted in cancer field and include following points as well.
1. Authors talks about microRNA that repress EZH2. What all microRNA's are known to do this, explain in detail.
2. There is no discussion about shear stress and EZH2 since shear stress is another key factor atherosclerosis.
3. There is no discussion about GAS5 and EZH2 interaction.
Please include a endothelial cell signaling diagram for EZH2 and SIRT1.
Author Response
Reviewer 1:
Comments and Suggestions for Authors
Manuscript written by Jolien Fledderus e.al., talks about the role of two epigenetic enzymes; EZH2 and SIRT1. The article is written nicely. However, authors have given less attention to EZH2 and SIRT1. Half of the article is talking about basic of atherosclerosis in detail. Only sections; 4 talks about these two epigenetic enzymes. My major comments on the article are;
The section 1 and 2 can be reduced since general process of atherosclerosis is already known and well-defined.
We thank the reviewer for this comment. We acknowledge the suggestion by the reviewer and agree that the general process of atherosclerosis is well-established within the cardiovascular field. However, we believe that providing this background information is important to also make the manuscript appealing to a more general audience, including junior scientists. Besides, detailing how endothelial cell (dys)function influences atherogenesis at different stages is the common structure of our review. We believe it is important to maintain this information to help the reader understand the potential of EZH2 and SIRT1 as anti-atherogenic therapy. Hence, we chose not to reduce the word count of sections 1 and 2.
In section 4, role of SIRT1 is well established in cardiovascular complications. Break this section into two epigenetic enzymes as sub-sections. They cannot be discussed together. Discuss more about EZH2 since its mostly targeted in cancer field and include following points as well.
We thank the reviewer for this valuable suggestion. In our original submission, we combined the discussion on these two epigenetic enzymes, because we thought this would be the clearest way to present this information. However, we agree with the reviewer that discussing the two epigenetic enzymes separately has its merits and therefore in our revised manuscript, we separated the discussion on EZH2 and SIRT1. In line with the reviewers suggestion, we extended the section on EZH2 with analogue examples from the cancer field (page 9-10) and discuss the observations in the context of atherosclerosis. Added references are:
Cao, Q.; Yu, J.; Dhanasekaran, S.M.; Kim, J.H.; Mani, R.S.; Tomlins, S.A.; Mehra, R.; Laxman, B.; Cao, X.;
Yu, J., et al. Repression of E-cadherin by the polycomb group protein EZH2 in cancer. Oncogene 2008, 27, 7274-7284, doi:10.1038/onc.2008.333.
Roche, J. The Epithelial-to-Mesenchymal Transition in Cancer. Cancers (Basel) 2018, 10,
doi:10.3390/cancers10020052.
Luo, H.; Jiang, Y.; Ma, S.; Chang, H.; Yi, C.; Cao, H.; Gao, Y.; Guo, H.; Hou, J.; Yan, J., et al. EZH2
promotes invasion and metastasis of laryngeal squamous cells carcinoma via epithelial-mesenchymal transition through H3K27me3. Biochemical and biophysical research communications 2016, 479, 253-259, doi:10.1016/j.bbrc.2016.09.055.
Zhao, M.; Hu, X.; Xu, Y.; Wu, C.; Chen, J.; Ren, Y.; Kong, L.; Sun, S.; Zhang, L.; Jin, R., et al. Targeting of
EZH2 inhibits epithelial‑mesenchymal transition in head and neck squamous cell carcinoma via regulating the STAT3/VEGFR2 axis. Int J Oncol 2019, 55, 1165-1175, doi:10.3892/ijo.2019.4880.
- Authors talks about microRNA that repress EZH2. What all microRNA's are known to do this, explain in detail.
We thank the reviewer for this question. Indeed, we elaborate on the effect of microRNAs that repress EZH2, but refrain from naming specific microRNAs. Indeed, multiple microRNAs are reported to target EZH2, and discussing all in detail is beyond the scope of our manuscript. Nevertheless, in our revised manuscript we discuss a few well-established microRNAs that are involved in the repression of EZH2 and discuss their relationship with atherosclerotic risk factors (page 10, line 8-15). Literature references included are:
Smits, M.; Mir, S.E.; Nilsson, R.J.; van der Stoop, P.M.; Niers, J.M.; Marquez, V.E.; Cloos, J.;
Breakefield, X.O.; Krichevsky, A.M.; Noske, D.P., et al. Down-regulation of miR-101 in endothelial cells promotes blood vessel formation through reduced repression of EZH2. PloS one 2011, 6, e16282, doi:10.1371/journal.pone.0016282.
Italiano, A. Role of the EZH2 histone methyltransferase as a therapeutic target in cancer. Pharmacol
Ther 2016, 165, 26-31, doi:10.1016/j.pharmthera.2016.05.003.
Vella, S.; Pomella, S.; Leoncini, P.P.; Colletti, M.; Conti, B.; Marquez, V.E.; Strillacci, A.; Roma, J.;
Gallego, S.; Milano, G.M., et al. MicroRNA-101 is repressed by EZH2 and its restoration inhibits tumorigenic features in embryonal rhabdomyosarcoma. Clinical epigenetics 2015, 7, 82, doi:10.1186/s13148-015-0107-z.
- There is no discussion about shear stress and EZH2 since shear stress is another key factor atherosclerosis.
We fully agree with the reviewer that shear stress is an important determinant of the susceptibility to develop atherosclerosis. In our revised manuscript, we emphasize this view by adding discussion on this topic to the introduction (page 1, lines 38-41) and to section 4 (page 9 lines 19-21). More precisely, we discuss the influence of shear stress on the expression of EZH2, which may be part of the atheroprotective mechanism that shear stress evokes. Additionally, table 2 is updated according to the text. References included are:
Lee, E.S.; Boldo, L.S.; Fernandez, B.O.; Feelisch, M.; Harmsen, M.C. Suppression of TAK1 pathway by
shear stress counteracts the inflammatory endothelial cell phenotype induced by oxidative stress and TGF-β1. Scientific reports 2017, 7, 42487, doi:10.1038/srep42487.
van Thienen, J.V.; Fledderus, J.O.; Dekker, R.J.; Rohlena, J.; van Ijzendoorn, G.A.; Kootstra, N.A.;
Pannekoek, H.; Horrevoets, A.J. Shear stress sustains atheroprotective endothelial KLF2 expression more potently than statins through mRNA stabilization. Cardiovascular research 2006, 72, 231-240, doi:10.1016/j.cardiores.2006.07.008.
- There is no discussion about GAS5 and EZH2 interaction.
The reviewer is correct. We did not include information on the interaction between GAS5 and EZH2 because this interaction is endothelial-cell independent and the primary focus of our review lies on endothelial cell dependent interactions in atherogenesis. However, we agree with the reviewer that the mention of non-endothelial repression of EZH2 may be beneficial in atherosclerosis and should be included for completeness, we shortly touch upon endothelial cell-independent anti-atherogenic effects of EZH2 and SIRT1. In our revised manuscript, we now also included the information on the interaction of GAS5 and EZH2 (page 11, line 12-13).
Please include a endothelial cell signaling diagram for EZH2 and SIRT1.
We thank the reviewer for this valuable suggestion and have included a signaling diagram that visualizes the endothelial signaling mechanisms by which EZH2 and SIRT1 influence atherogenesis (figure 3)
Reviewer 2 Report
Overall, this is a succinct and well-written synopsis of atherosclerotic disease and treatments, existing and potential. Some content and arguments could be made clearer for the reader.
1) First paragraph, what is "cardiac stroke"...are you referring to heart attack or stroke in general?
2) Some statements are awkward or largely uninterpretable as stated:
Pg 2, first sentence...Pg 3, next to last sentence...Pg 11, first paragraph...Also, check for typos throughout.
3) Fig 1, Text is too small...some acronyms are not spelled out/defined in fig legend. In last panel of C, what is indicated by the factors listed in the rectangle (up/down)? While it is unnecessary to include everything here, there should be enough information in the fig (and/or legend) to understand the concept.
4) Table 1 and text, why are angiotensin receptor blockers (ARBs) not included as hypertensive treatments?
Author Response
Reviewer 2:
Comments and Suggestions for Authors
Overall, this is a succinct and well-written synopsis of atherosclerotic disease and treatments, existing and potential. Some content and arguments could be made clearer for the reader.
- First paragraph, what is "cardiac stroke"...are you referring to heart attack or stroke in general?
We thank the reviewer for this critical remark. Indeed, cardiac stroke is an imprecise term. We are referring to stroke in general as one of the pathologies that may stem from atherosclerosis. We have changed the text from “cardiac stroke” to “stroke”, to make this more clear.
- Some statements are awkward or largely uninterpretable as stated: Pg 2, first sentence...Pg 3, next to last sentence...Pg 11, first paragraph... Also, check for typos throughout.
We have reassessed the text fragments mentioned by the reviewer and made adjustments where necessary. We carefully proof-read our revised manuscript and corrected typos throughout.
- Fig 1, Text is too small...some acronyms are not spelled out/defined in fig legend. In last panel of C, what is indicated by the factors listed in the rectangle (up/down)? While it is unnecessary to include everything here, there should be enough information in the fig (and/or legend) to understand the concept.
Text of figure 1 is too small:
In our revised manuscript we adjusted the text size of figure 1.
Some acronyms are not spelled out/defined in fig. legend:
Some acronyms are not spelled out/defined in the fig. legend because they are mostly mentioned in the corresponding text. However, because it makes it more clear for the reader we changed this and defined the acronyms again in the figure legend (page 3, fig 1).
What is indicated by the factors listed in the rectangle (fig 1. C3)
We thank the reviewer for this question. The factors listed in the rectangle in panel C3 of figure 1 represents the loss of endothelial cell markers (factors in pink) and an increase in mesenchymal markers (factors in brown) during the process of Endothelial-Mesenchymal transition. Following the reviewers comment, we agree that this should be more clearly indicated and mentioned in the figure legend. In the revised figure, we moved the rectangle to panel B3 and adjusted the figure legend accordingly.
- Table 1 and text, why are angiotensin receptor blockers (ARBs) not included as hypertensive treatments?
We thank the reviewer for this critical remark. We omitted ARBs as hypertensive treatment because, to the best of our knowledge, ARBs are not included in standard treatment guidelines of hypertension during atherosclerosis (1-3). However, we do acknowledge that ARBs are standard therapy for hypertension and therefore in our revised manuscript we included ARBs in table 1 (page 6) and added the missing references (ref. nr. 1 and 4).
- Piepoli MF, Hoes AW, Agewall S, Albus C, Brotons C, Catapano AL, et al. 2016 European Guidelines on cardiovascular disease prevention in clinical practice: The Sixth Joint Task Force of the European Society of Cardiology and Other Societies on Cardiovascular Disease Prevention in Clinical Practice (constituted by representatives of 10 societies and by invited experts)Developed with the special contribution of the European Association for Cardiovascular Prevention & Rehabilitation (EACPR). Eur Heart J. 2016;37(29):2315-81.
- Mach F, Baigent C, Catapano AL, Koskinas KC, Casula M, Badimon L, et al. 2019 ESC/EAS Guidelines for the management of dyslipidaemias: lipid modification to reduce cardiovascular risk. Eur Heart J. 2020;41(1):111-88.
- Arnett DK, Blumenthal RS, Albert MA, Buroker AB, Goldberger ZD, Hahn EJ, et al. 2019 ACC/AHA Guideline on the Primary Prevention of Cardiovascular Disease: A Report of the American College of Cardiology/American Heart Association Task Force on Clinical Practice Guidelines. Circulation. 2019;140(11):e596-e646.
- Dézsi, C.A. The Different Therapeutic Choices with ARBs. Which One to Give? When? Why? Am J Cardiovasc Drugs 2016, 16, 255-266, doi:10.1007/s40256-016-0165-4.
Round 2
Reviewer 1 Report
Authors have responded to my comments. I see a lot of spelling mistakes for example: Figure legend 3: homeodstasis (IInd line), inflammaroty (3rd line). Please correct these and additionally check again the entire manuscript for this.
Author Response
Comments and Suggestions for Authors
Authors have responded to my comments. I see a lot of spelling mistakes for example: Figure legend 3: homeodstasis (IInd line), inflammaroty (3rd line). Please correct these and additionally check again the entire manuscript for this.
We thank the reviewer for the comments and apologize for our omission to carefully check the revised manuscript prior to its submission. The spelling mistakes in figure legend 3, referenced by the reviewer, have been corrected (page 11). In addition, we have used the MS Editor package to check for any remaining errors in spelling or grammar, and made adjustments where needed. Last, we have changed the grammar used in the revised manuscript from US English to UK English, as per journal preference. A “track changes” manuscript has been uploaded in supplement for the reviewer’s convenience.